# Evaluation of Ability of Inactivated Biomasses of *Lacticaseibacillus rhamnosus* and *Saccharomyces cerevisiae* to Adsorb Aflatoxin B_1_ In Vitro

**DOI:** 10.3390/foods13203299

**Published:** 2024-10-17

**Authors:** Rogério Cury Pires, Julia da Costa Calumby, Roice Eliana Rosim, Rogério D’Antonio Pires, Aline Moreira Borowsky, Sher Ali, Esther Lima de Paiva, Ramon Silva, Tatiana Colombo Pimentel, Adriano Gomes da Cruz, Carlos Augusto Fernandes de Oliveira, Carlos Humberto Corassin

**Affiliations:** 1Departamento de Zootecnia, Escola Superior de Agricultura Luiz de Queiroz, Universidade de São Paulo, Av. Pádua Dias, 11, Piracicaba 13418-900, SP, Brazil; rogeriocury@usp.br (R.C.P.); dantonio@usp.br (R.D.P.); alineborowsky@usp.br (A.M.B.); 2Departamento de Engenharia de Alimentos, Faculdade de Zootecnia e Engenharia de Alimentos, Universidade de São Paulo, Av. Duque de Caxias Norte, 225, Pirassununga 13635-900, SP, Brazil; juliacosta@usp.br (J.d.C.C.); roice@usp.br (R.E.R.); alisher@usp.br (S.A.); elpaiva@usp.br (E.L.d.P.); carloscorassin@usp.br (C.H.C.); 3Instituto Federal do Paraná, R. Felipe Tequinha Street, 1400, Paranavaí 87703-536, PR, Brazil; ramonsg@uol.com.br (R.S.); tatipimentel@hotmail.com (T.C.P.); 4Departamento de Alimentos, Instituto Federal de Educação, Ciência e Tecnologia do Rio de Janeiro, R. Sen. Furtado, 121/125, Rio de Janeiro 20270-021, RJ, Brazil; adriano.cruz@ifrj.edu.br

**Keywords:** adsorption, AFB_1_, *L. rhamnosus*, *S. cerevisiae*, functional yogurt

## Abstract

Biological decontamination strategies using microorganisms to adsorb aflatoxins have shown promising results for reducing the dietary exposure to these contaminants. In this study, the ability of inactivated biomasses of *Lacticaseibacillus rhamnosus* (LRB) and *Saccharomyces cerevisiae* (SCB) incorporated alone or in combination into functional yogurts (FY) at 0.5–4.0% (*w*/*w*) to adsorb aflatoxin B_1_ (AFB_1_) was evaluated in vitro. Higher adsorption percentages (86.9–91.2%) were observed in FY containing 1.0% LR + SC or 2.0% SC (*w*/*w*). The survival of mouse embryonic fibroblasts increased after exposure to yogurts containing LC + SC at 1.0–4.0% (*w*/*w*). No significant differences were noted in the physicochemical and sensory characteristics between aflatoxin-free FY and control yogurts (no biomass) after 30 days of storage. The incorporation of combined LRB and SCB into yogurts as vehicles for these inactivated biomasses is a promising alternative for reducing the exposure to dietary AFB_1_. The results of this trial support further studies to develop practical applications aiming at the scalability of using the biomasses evaluated in functional foods to mitigate aflatoxin exposure.

## 1. Introduction

Aflatoxins are toxic secondary metabolites produced by certain fungi species belonging to the genus *Aspergillus*, which develop on crops, foods, and feeds [1,2]. To date, more than 20 types of aflatoxins have been identified, while aflatoxin B_1_ (AFB_1_) is considered the major toxic metabolite produced by fungi that can contaminate foods and feeds [3]. AFB_1_ is classified as a group 1 carcinogen by the International Agency for Research on Cancer [4]. Consistent with the type, dose, and time of exposure through contaminated food, AFB_1_ can affect the liver as the main target organ and cause immunosuppression, carcinogenic, mutagenic, and teratogenic effects [5].

Because of the high resistance of aflatoxins to food processing techniques, prevention of fungi infection and development on food materials is the main strategy to avoid aflatoxin contamination [3]. However, decontamination can be partially achieved using some physical, chemical, or biological approaches aiming at inactivating, degrading, or sequestrating the aflatoxin in the food matrix [6]. In this context, biological methods using microorganisms to adsorb aflatoxins have shown promising results for reducing these toxins in foods and feeds [7,8]. When microrganisms such as lactic acid bacteria (LAB) and some yeast species are added to a fermented food, they may act like “enterosorbents” and bind to the mycotoxins, thereby alleviating their bioavailability in the gastrointestinal tract [3]. Regarding the reduction or removal of AFB_1_, both *Lacticaseibacillus rhamnosus* and *Saccharomyces cerevisiae* have exhibited excellent performance [7,8]. Although the mechanisms behind the adsorption are not completely understood, there is evidence of the formation of a reversible, stable complex of mycotoxin on the surface of subjected microorganisms, without any chemical modifications, with similar adsorption capacities shown by viable or inactivated microorganisms [8].

Information and consumers’ awareness regarding health has led to the search for a healthier lifestyle, a fact that has increased the demand for functional foods containing probiotics. Probiotics are defined as live microorganisms that are intended to have health benefits when consumed or applied to the body [9]. Probioticity is a strain-specific characteristic and most probiotics belong to the metabolic group of lactic fermentation bacteria such as many *Lactobacillus* species [10], although recent investigations have presented some yeast species with probiotic properties, including *Saccharomyces boulardii* (reviewed by Souza et al. [11]). The use of probiotics is valuable in reducing several health risks through its incorporation into functional foods, as they can vitally improve public health, especially in critical age groups [9]. The choice of a suitable food product for delivering probiotics depends on the type of probiotic strain, the food’s ability to protect the probiotics, and consumer preferences. In this context, yogurt is one of the most common probiotic-enriched foods, mainly because its matrix acts as a buffer against stomach acid, helping probiotics reach the gut alive [10]. However, probiotic supplementation for fermented products like yogurt can potentially alter its sensory properties [12]. This stresses the demand for examining the sensory features of functional products containing probiotics, aiming at increasing its acceptability among consumers [13].

A possible alternative for adding beneficial microorganisms to food products is the use of paraprobiotics, defined as inactivated microbial cells (non-viable) that provide a health benefit to the consumer [14]. There is evidence on the anti-inflammatory effect and positive immune responses of paraprobiotics in animals and humans, with some advantages if compared with probiotics since non-viable microbial cells may exhibit enhanced safety and technological advantages, such as longer shelf life and low interference with formulations of the food matrix [15]. These features are in line with the scientific advances on the use of microbial products as a promising approach in the decontamination of mycotoxins [16]. In these products, the use of paraprobiotics and inactivated cells of other, non-probiotic microorganisms is an efficient approach to adsorb toxins in a target matrix of a food or beverage. Abdel-Salam et al. [17] demonstrated the efficacy of a functional yogurt containing viable cells of *Lactobacillus acidophilus*, *Streptococcus thermophilus*, and *Bifidobacterium bifidum* to protect against AFB_1_ toxicity in rats. However, to the authors’ knowledge, there is no information available on the protective effects of functional yogurts containing inactivated biomasses of microorganisms against the cytotoxic effects of AFB_1_, or on the potential influence of the incorporation of these products on the quality parameters of yogurts. Therefore, this study aimed to evaluate the efficacy of inactivated biomasses of the probiotic LAB *L. rhamnosus* and *S. cerevisiae*, a non-probiotic yeast, incorporated alone or in combination into plain, non-flavored yogurts [18] as vehicles for these inactivated microorganisms, to adsorb AFB_1_ in vitro and reduce its toxicity to mouse embryonic fibroblasts (MEF-1). The effect of the incorporation of biomasses on the overall quality of the experimental yogurts was also assessed.

## 2. Materials and Methods

### 2.1. Preparation of Lactic Acid Bacteria and Yeast Biomasses

In this study, a commercially available lyophilized *L. rhamnosus* (HOWARU^®^ LYO 40 DCU, Danisco Ltd., São Paulo, Brazil) biomass containing 1.0 × 10^10^ viable cells/g was used. This strain was previously evaluated regarding its ability to bind to AFB_1_ in phosphate buffer solution (PBS), with a percentage binding of 45.9% at pH 3.0 [19]. The *S. cerevisiae* strain was a commercially available brewer’s biological dry yeast biomass (Fermentis K-97, SafAle, Bruggeman, Gent, Belgium) containing 1.0 × 10^10^ viable cells/g. The AFB_1_ adsorption capacity of the *S. cerevisiae* strain in PBS was 45.5% to bind to AFB_1_ in PBS at pH 3.0 [20]. Both the *L. rhamnosus* strain in the lyophilized starter culture and *S. cerevisiae* cells were inactivated by autoclaving the material at 121 °C for 10 min. Heat inactivation of microbial cells may increase their aflatoxin binding properties, due to denaturation of membrane proteins, peptidoglycans, and degradation of polysaccharide components of the cell wall, which ultimately change the hydrophobicity of the cell wall and respective binding capacities [19,21]. After this procedure, the inactivated biomasses of the *L. rhamnosus* (LRB) and *S. cerevisiae* (SCB) strains were stored at −20 °C, until incorporation into plain yogurts.

### 2.2. Preparation of the Functional Yogurts

Plain stirred yogurt containing LRB and SCB, alone or in combination, was produced following the procedures described by Soukoulis et al. [19], with some modifications. Briefly, 150 L of unpasteurized skim milk containing <0.1% fat was obtained from a local dairy plant and standardized by adding 5.3 kg of skim milk powder to achieve a total solids content (TS) of 12.5% (*w*/*w*). After standardization, duplicate 100 mL samples were collected and analyzed using a Lactoscan ^®^ MCC (Milkotronic, Nova Zagora, Bulgaria) for confirmation of TS and to determine the percentages of fat, total protein, lactose, and minerals. Furthermore, triplicate samples of the standardized milk were analyzed according to Jager et al. [22] and found to have AFB_1_ or AFM_1_, the hydroxilated metabolite of AFB_1_, at levels below the detection limits of the analytical method (0.1 and 0.075 µg/mL, respectively). Next, aliquots of the standardized milk were assigned to 13 yogurt-producing vats (15 L per vat), then incorporated with increasing percentages (*w*/*w*) of LRB or SCB biomasses, as follows: Vat 1: no LRB or SCB (control); Vat 2: 0.5% LRB; Vat 3: 0.5% SCB; Vat 4: 0.5% LRB + 0.5% SCB; Vat 5: 1.0% LRB; Vat 6: 1.0% SCB; Vat 7: 1.0% LRB + 1.0% SCB; Vat 8: 2.0% LRB; Vat 9: 2.0% SCB; Vat 10: 2.0% LRB + 2.0% SCB; Vat 11: 4.0% LRB; Vat 12: 4.0% SCB; Vat 13: 4.0% LRB + 4.0% SCB. The contents in all vats were thoroughly mixed using a manual milk mixer during 10 min.

The mixtures were heat-treated at 90 °C for 15 min, then cooled to 42 °C for inoculation of the starter culture (3% inoculum) containing *Streptococcus thermophilus* and *L. delbrueckii* ssp. *bulgaricus* (Yo-Flex, Chr. Hansen, Horsholm, Denmark). The vats were incubated at 42 °C for approximately 3 h, until the products reached a pH of 4.5. The obtained functional yogurts (FY) were cooled rapidly to 10 °C, transferred to 1 L polyethylene bottles, and stored at 5 °C for 30 days.

### 2.3. Adsorption Assays of Aflatoxin B_1_ in Functional Yogurts In Vitro

The AFB_1_ binding assays were conducted using the protocol described by Bovo et al. [19], with some modifications. A working solution of AFB_1_ at 10.0 µg/mL was prepared in PBS at pH 3.0, and 1 mL aliquots of this solution were transferred to 15 mL test tubes. After this procedure, 10 g of each functional yoghurt containing LRB, SCB, or LRB + SCB at 0.5–4.0% (*w*/*w*) and control (yogurt without any biomass incorporation) was added to triplicate tubes and the mixtures were vortexed for 5 min. The concentration of AFB_1_ in the final mixtures in all tubes was 1.0 µg/g. The tubes were placed on a rotating shaker at 180 rpm for 60 min at room temperature (25 °C). Next, the mixtures were centrifuged at 3100× *g* for 10 min, and the supernatants were separated for quantification of AFB_1_.

AFB_1_ in the supernatants was extracted and purified using immunoaffinity columns (Aflatest WB^®^, Vicam, Watertown, MA, USA), exactly as described by Jager et al. [22]. For derivatization of AFB_1_ in samples, 50 µL of the final extract was diluted in 5 mL of milli-Q water (Millipore, Burlington, MA, USA), and 100 µL of the resulting solution was mixed with 900 µL of acetonitrile/water (50:50) in an Eppendorf tube. Then, 500 µL was transferred to a new Eppendorf tube containing 200 µL hexane and 100 µL trifluoroacetic acid, kept at 35 °C for 10 min, evaporated to near-dryness and diluted in 500 µL acetonitrile/water (50:50). Twenty microliters of the final extracts was injected into a Shimadzu 10VP liquid chromatograph (Kyoto, Japan), equipped with a 10 AXL fluorescence detector (excitation at 360 nm and emission above 440 nm). The chromatographic run was achieved using a Kinetex C18 column (Phenomenex, Torrance, CA, USA) 4.6 × 150 mm, 2.6 μm particle size, and the isocratic mobile phase consisted of methanol/water/acetonitrile (61.4:28.1:10.5, *v*/*v*/*v*) with a flow rate of 0.50 mL/min. Five-point calibration curves containing the AFB_1_ standard diluted in acetonitrile and derivatized as previously described for the samples were prepared at levels of 0.1, 0.25, 0.5, 0.75, and 1.0 μg/mL. Integrated peak areas were linearly correlated with the concentrations. Identification of AFB_1_ was achieved by comparing the retention time of AFB_1_ peaks in the samples with the standards in the calibration curves. The limits of detection (LOD) and limits of quantification (LOQ) were calculated at a signal-to-noise ratio of 3 and 10, respectively, being 0.017 and 0.055 μg/kg. The analytical method was previously validated with contaminated yogurt samples at levels of 0.2 and 0.5 μg/kg (*n* = 3, for each concentration), which resulted in recovery rates from 72 to 93% [22].

The percentage of mycotoxin binding was calculated using Equation (1), in which “a” indicates the percentage of AFB_1_ adsorbed by LRB and/or SCB, “b” is the concentration of AFB_1_ added to buffer (1.0 µg/mL), “c” is the concentration of AFB_1_ in the buffer solution plus yeast and/or LAB inactivated cells after centrifugation, and “d” is the concentration of any interferences in the negative control (buffer solution + LRB/SCB).
a = [b − (c − d)]/b × 100(1)

### 2.4. Survival of Mouse Embryonic Fibroblasts Exposed to Aflatoxin B_1_ in Functional Yogurts

Procedures for assessing the cell viability of MEF-1 to AFB_1_ in FY incorporated with both LRB and SCB were performed as described by Nones et al. [23], with some modifications. MEF-1 cells derived from the American Tissue Culture Collection (ATCC) were obtained from the Cell Bank of Rio de Janeiro (BCRJ) and cultured in Dulbecco’s modified Eagle medium (DMEM) Gibco^®^ (Thermo Fischer Scientific, Waltham, MA, USA) supplemented with 10% (*v*/*v*) fetal bovine serum, penicillin (100 unit/mL), and streptomycin (100 µg/mL). MEF-1 cells were reseeded after trypsination on a weekly basis in a 1:5 split ratio and allowed to grow as monolayers in cell culture flasks (75 cm^2^) with filter screw caps (Corning^®^, Corning, MA, USA) until reaching 90–100% confluency. MEF-1 cells from passages between 14 and 18 were maintained at 37 °C in a humidified atmosphere of 5% CO_2_ and reserved for cell viability experiments.

MEF-1 cell viability was assessed using the 3-{4,5-dimethylthiazol-2-yl} diphenyltetrazolium bromide (MTT) assay (Sigma-Aldrich, St. Louis, MO, USA), which measures the cellular metabolic activity as an indicator of cell viability, proliferation, and cytotoxicity [24]. MTT stock solution was prepared in phosphate-buffered saline at 5.0 mg/mL, and the working solution was prepared in DMEM at 0.5 mg/mL. MEF-1 cells were seeded in triplicate in 96-well culture plates at 12,000 cells/cm^2^ and incubated overnight at 37 °C with 5% CO_2_ in supplemented DMEM. On the following day, the medium was replaced with 200 µL of one of the following treatment media: (1) DMEM only (control); (2) DMEM + AFB_1_ at 1.0 µg/mL; (3) DMEM + FY without any LRB or SCB; (4) DMEM + FY without any LRB or SCB + AFB_1_ (1.0 µg/mL); (5) DMEM + FY with 0.5% (*v*/*v*) of LRB and SCB; (6) DMEM + FY with 0.5% (*v*/*v*) of LRB and SCB + AFB_1_ (1.0 µg/mL); (7) DMEM + FY with 1.0% (*v*/*v*) of LRB and SCB; (8) DMEM + FY with 1.0% (*v*/*v*) of LRB and SCB + AFB_1_ (1.0 µg/mL); (9) DMEM + FY with 2.0% (*v*/*v*) of LRB and SCB; (10) DMEM + FY with 2.0% (*v*/*v*) of LRB and SCB + AFB_1_ (1.0 µg/mL); (11) DMEM + FY with 4.0% (*v*/*v*) of LRB and SCB; (12) DMEM + FY with 4.0% (*v*/*v*) of LRB and SCB + AFB_1_ (1.0 µg/mL).

The plates were incubated for 24 h, then the treatment medium was gently replaced with 100 µL fresh medium containing 10% (*v*/*v*) of MTT stock solution. After incubation for 4 h at 37 °C, the medium was replaced by dimethylsulphoxide (DMSO) and plates shaken to dissolve the purple formazan product. The absorbance of each well was read at 570 nm using an EZ Read 2000^®^ plate reader (Biochrom, Cambridge, UK). Cell viability was expressed in percentages, according to Equation (2). The experiments were performed in triplicate and each result represents the mean of at least three independent experiments.
Cell viability = [Absorbance_570_ Treatments/Absorbance_570_ Control] × 100 (2)

### 2.5. Physicochemical and Sensory Evaluation of Functional Yogurts

FY samples were collected and analyzed on d 30 of storage at 5 °C, to assess the possible effects of LRB and/or SCB on the physicochemical characteristics and sensory grades of the products. Fat, protein, and pH were determined according to the Association of Official Analytical Chemists [25]. For sensory evaluation, samples of FY without any aflatoxin were submitted to a trained panel of 15 individuals (8 men and 7 women, aged 18–25 years), according to Mousavi et al. [26]. This study was approved by the Research Ethics Committee of the FZEA-USP (approval no. 5.742.458). Panelists were trained during 3 sessions, one per week for 3 weeks, to score intensity of appearance, consistency, aroma, and taste, using samples of commercial skim, plain yogurts with TS of 11.5% (*w*/*w*) and without any LRB or SCB as reference material. Definitions of sensory attributes related to the typical characteristics of yogurts were based on criteria described by Aktar [27], as follows: appearance: white coagulum, uniform; consistency: moderate viscosity, homogeneous; aroma and taste: pure, typically acid. All attributes were quantified by panelists according to an intensity scale from 1 to 5, in which a rating score of 5 was equivalent to the typical characteristics of yogurt used as reference material and full presence of attributes; 1 was equivalent to a product without typical characteristics and non-detectable attributes.

### 2.6. Statistical Analysis

The results were subjected to analysis of variance, in accordance with the general linear model (GLM) of SAS^®^ [28], to check for significant differences between mean values. When applicable, the Fisher LSD test (least significant difference) was used for comparison between mean values, considering *p* < 0.05.

## 3. Results and Discussion

Table 1 presents the levels of AFB_1_ and the respective reduction percentages by FY containing LRB, SCB, or their combinations. Lower levels of AFB_1_ (*p* < 0.05) were observed in tubes with FY containing the combined biomasses of inactivated cells (LRB + SCB). Consequently, the highest adsorption percentages (*p* < 0.05) were observed with inclusions of 1.0–4.0% of both biomasses, in which the AFB_1_ reduction ranged from 86.9 to 91.2%. The second-best response was attained with FY containing 1.0–4.0% of SCB alone, with respective AFB_1_ reduction values ranging from 81.8 to 84.8%. FY containing only LRB provided AFB_1_ reduction rates of 11.3 to 42.8%. The mechanisms involved in the adsorption of AFB_1_ by microbial cells, either viable or inactivated, are not fully understood. However, the adsorption process may involve physical and chemical interactions, such as aflatoxin binding to cell wall components including polysaccharides, proteins, and lipids in LAB cells [29], and β-glucans, mannoproteins, and chitin in yeast cells [30]. β-glucans, in particular, have been shown to adsorb aflatoxins via hydrogen bonding and other interactions [31]. In addition, hydrophobic and electrostatic interactions between the AFB_1_ molecule and the lipid-rich regions of the microbial cell wall, and weak, non-covalent interactions such as van der Waals forces between AFB_1_ and the microbial cell surface also contribute to the adsorption process [29,30].

Comparing the adsorption results obtained in this work with those described in other studies is a difficult task, due to the absence of previous data on the removal of aflatoxins by combinations of LRB and SCB conveyed in yogurts. However, our findings are consistent with those reported by Zolfaghari et al. [32], who observed that both bacterial and yeast isolates had the capacity to effectively reduce the levels of AFB_1_. The authors observed that *Lactobacillus* spp. exhibited a binding capacity ranging from 8.4 to 31.1%, while *S. cerevisiae* achieved an AFB_1_ binding rate of 30.5%. Data presented in this work align with previous findings regarding the AFB_1_ removal capacity of *L. rhamnosus* [19]. Our results are also in agreement with data reported in previous studies, such as El-Nezami et al. [33] and Luo et al. [34], in which *L. rhamnosus* demonstrated similar capacities to bind to AFM_1_ in whole milk, with binding percentages ranging from 36 to 63%. Moreover, Pourmohammadi et al. [35] demonstrated that the combined use of LAB strains enhances the adsorption rate of mycotoxins, when compared with the isolated binding capacities of these strains, which corroborates the improved adsorption results achieved with the LRB and SCB in our study. These findings collectively underscore the potential of both *S. cerevisiae* and LAB strains for reducing the aflatoxin contamination in foods, holding promise for further exploration and industrial applications.

The survival of MEF-1 cells after treatment with AFB_1_ and/or yogurts incorporated with different percentages of LRB + SCB is presented in Figure 1. The significant increases in cell survival after exposure to yogurts containing ≥1% LRB + SCB indicate a potential protective effect against AFB_1_-induced cytotoxicity. This suggests that the incorporation of LRB and SCB into yogurt formulations may mitigate the adverse effects of AFB_1_ exposure, highlighting the potential application of paraprobiotics and other inactivated non-probiotic microorganisms in food safety and health promotion. Although the reversibility of the AFB_1_ adsorption by LRB or SCB was not assessed in the present experiment, the higher survival of MEF-1 cells indicates a lower bioavailability of AFB_1_ in yogurts containing ≥1% LRB + SCB after 24 h exposure.

Similar protective effects of probiotic microorganisms against aflatoxins were reported by Martinez et al. [36], regarding the adsorption and degradation of AFM_1_ by selected species of LAB and yeasts in fluid milk. The authors assessed the ability of tested probiotics to detoxify AFM_1_ using brine shrimp (*Artemia salina*) toxicity assays and found that probiotics *Pediococcus pentosaceus* and *Kluyveromyces marxianus* have the ability to adsorb and degrade AFM_1_ to less toxic metabolites in milk. Regarding AFB_1_, its cytotoxicity has been extensively demonstrated in several cell cultures, such as a human liver cancer cell line (HepG2), the human embryonic kidney 293 cell line (HEK293T), and continuous porcine cell lines from alveolar macrophages (3D4/21) [37]. Conversely, cell cultures are practical and viable models for studies on mycotoxin detoxification. The protective effects of a modified bentonite clay against the toxic effects of AFB_1_ in fibroblasts 3T3 and epithelial colorectal adenocarcinoma cells (Caco-2) were reported by Nones et al. [23].

In this study, the MEF-1 cells were used for the first time to highlight the protective effects of LRB and SCB incorporated into FY against the AFB_1_-induced cytotoxicity. A notable advantage of using MEF-1 cells is their ease of collection and culture, making them readily accessible for experimentation. Additionally, MEF-1 cells exhibit robust growth and survival characteristics, providing a reliable and reproducible cell model for toxicity assessments [38]. Their ability to support embryonic stem cell (ESC) pluripotency and enhance plating efficiency makes them particularly suitable for studying developmental toxicity and cellular differentiation processes [39]. Furthermore, previous studies have used MEFs as an “embryonic” model for better understanding the mechanism of action for other mycotoxins, such as fumonisin B_1_ [40].

The physicochemical characteristics of the FY without aflatoxins but containing LRB, SCB, or their combinations at percentage inclusions of 0.5–4.0% (*w*/*w*) are presented in Table 2. No significant differences (*p* > 0.05) were observed in the evaluated physicochemical parameters (fat, protein, and pH) of FY containing cell biomass during 30 d storage. There are no available data on the physicochemical parameters of yogurts incorporated with LRB in combination with SCB. However, our results are consistent with previously reported data [41], indicating that the incorporation of viable probiotics at variable levels did not affect the physicochemical characteristics of the resulting yogurts.

Concerning the sensory evaluation of the FY after 30 days of storage, the results are shown in Table 3. As observed for physicochemical parameters (Table 3), no differences (*p* > 0.05) were found in the mean values scored for intensity of appearance, consistency, aroma, and flavor among yogurts containing 0.5–4.0% LRB, SCB, or LRB + SCB inclusions, when compared with controls (no cell biomass incorporated). It is noteworthy that the addition of certain ingredients, such as rice bran and fiber, has been shown to decrease the sensory scores of yogurts [42,43,44], which contrasts with the findings described in this work on the incorporation of inactivated cell biomasses of *L. rhamnosus* or *S. cerevisiae*. Despite the absence of literature data on the possible effects of inactivated biomasses of *L. rhamnosus* or *S. cerevisiae* on the sensory attributes of plain stirred yogurts, the incorporation of LRB and/or SCB did not significantly affect the acceptance of the FY produced in our study. Conversely, regarding the incorporation of living cells of probiotics into yogurts, some data indicated significantly higher scores in probiotic-enriched samples, thus suggesting a positive contribution of probiotic bacteria to the sensory attributes of the final product [45].

*L. rhamnosus* strains are considered as generally recognized as safe (GRAS) by the Food and Drug Administration (FDA) and have the qualified presumption of safety (QPS) status for intentional addition to food and feed granted by the European Food Safety Authority (EFSA) [46]. Due to its long history of safe use and consumption, most strains of *S. cerevisiae* are also classified as GRAS [47] for use in foods. In addition, the consumption of heat-inactivated *S. cerevisiae* strains administered in capsules or tablets was reviewed by Almada et al. [48], who highlighted their health benefits when administered with diet. This evidence confirms the safety of using the LRB and SCB biomasses as potential adsorbents for dietary aflatoxins.

In our work, the ability of a combination of inactivated microorganisms conveyed in yogurt to adsorb AFB_1_ in vitro was evaluated. Most of the studies using yogurt as a vehicle for beneficial microorganisms were performed with live isolated cells [10,17], which may change the physicochemical and sensory characteristics of the products. The utilization of inactivated microbial cells has been studied mainly using single or combined microorganisms in PBS, not incorporated into food matrices. Hence, the results presented here indicate for the first time that the addition of LRB and SCB to yogurts was effective in adsorbing AFB_1_ and reducing its toxicity to MEF-1 cells. In addition, the inclusion of biomasses in the FY had no influence on the physicochemical and sensory characteristics of the products, which also highlights the novelty of this study.

## 4. Conclusions

In this trial, LRB and/or SCB incorporated into yogurts effectively adsorbed AFB_1_ in vitro, with the highest adsorption percentages (86.9 to 91.2%) found with inclusions of 1.0–4.0% of both biomasses. In addition, the cytotoxicity of AFB_1_ to MEF-1 significantly reduced when the cells were simultaneously exposed to FY containing ≥1.0% (*w*/*w*) of LRB + SCB. Compared with controls (no biomass included), FY containing LRB and/or SCB at inclusion percentages of up to 4.0% (*w*/*w*) had no significant differences in the physicochemical characteristics and sensory attributes during 30 d of storage. Thus, the incorporation of combined LRB and SCB into yogurts as vehicles for these inactivated biomasses is a promising alternative for reducing the AFB_1_ toxic effects. However, the main limitations of this study involve the lack of in vivo data on the AFB_1_ adsorption ability of FY containing LRB and SCB. In particular, the protective effects of LRB and SCB incorporated into FY against the AFB_1_-induced MEF-1 cytotoxicity do not fully replicate the complex interactions and environments found in living organisms. In addition, in vitro systems do not account for metabolic processes that occur in the body, such as the conversion of substances into active or toxic metabolites in the liver, or cellular responses to toxic substances that may involve multi-step processes, interactions with other cells, or systemic effects. Finally, the concentration of AFB_1_ used in MEF-1 cell cultures in this study may not represent the usual levels of human exposure to dietary aflatoxins Therefore, to extend this research, future studies should explore in vivo experiments to assess the effectiveness of AFB_1_ adsorption by the evaluated FY under real conditions of aflatoxin exposure. Follow-up experiments in this direction could use animal models intoxicated with AFB_1_ to assess the protective effects of the ingested FY in the gastrointestinal tract against the toxic effects of this mycotoxin, as well as to ensure the absence of undesirable interactions of the FY with nutrients in the diet. This would provide further valuable information regarding the potential of the FY to serve as a preventive strategy for dietary exposure to aflatoxins.

## Figures and Tables

**Figure 1 foods-13-03299-f001:**
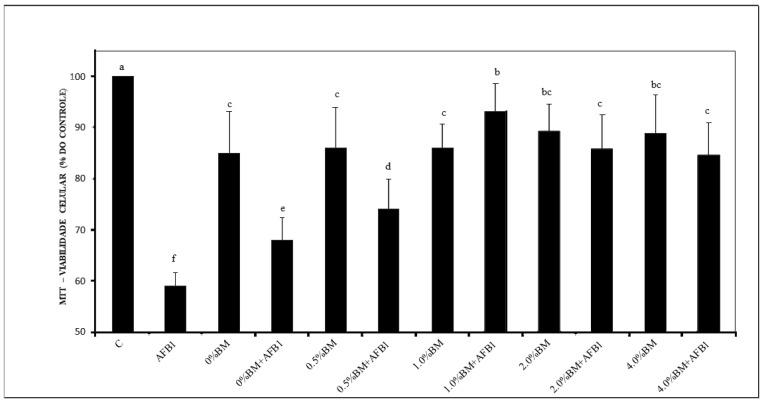
Survival of mouse embryonic fibroblasts (MEF-1) analyzed by the MTT assay, after exposure to 1.0 µg/mL of aflatoxin B_1_ (AFB_1_) and functional yogurts (FY) containing 0, 1.0, 2.0, or 4.0% (*w*/*w*) of cell inactivated biomasses (BM) of *Lacticaseibacillus rhamnosus* (1.0 × 10^10^ cells/g) and *Saccharomyces cerevisiae* (1.0 × 10^10^ cells/g). BM was inactivated by autoclaving at 121 °C for 10 min. Values are expressed as mean ± standard deviation of percentages relative to control (C, no exposure to AFB_1_ or FY containing BM) of 3 independent experiments with 3 replicates each. ^a–f^ Bars with different letters differ significantly (*p* < 0.05).

**Table 1 foods-13-03299-t001:** Aflatoxin B_1_ concentrations and reduction percentages in test tubes with functional yogurts stored at 5 °C for 30 days, containing different percentages of *Lacticaseibacillus rhamnosus* and *Saccharomyces cerevisiae* biomasses ^1^.

Treatment (Vat)	*L. rhamnosus* (%)	*S. cerevisiae* (%)	Aflatoxin B_1_ Level (µg/g) ^2^	Reduction (%) ^3^
1 (Control)	0	0	0.978 ± 0.022 ^a^	-
2	0.5	0	0.897 ± 0.009 ^b^	11.3 ^f^
3	0	0.5	0.406 ± 0.008 ^d^	60.4 ^c^
4	0.5	0.5	0.357 ± 0.010 ^d^	65.3 ^c^
5	1.0	0	0.782 ± 0.011 ^b^	22.8 ^e^
6	0	1.0	0.147 ± 0.014 ^e^	84.8 ^b^
7	1.0	1.0	0.098 ± 0.006 ^f^	91.2 ^a^
8	2.0	0	0.723 ± 0.014 ^b^	28.7 ^e^
9	0	2.0	0.168 ± 0.009 ^f^	81.8 ^b^
10	2.0	2.0	0.125 ± 0.009 ^ef^	88.2 ^ab^
11	4.0	0	0.582 ± 0.011 ^c^	42.8 ^d^
12	0	4.0	0.127 ± 0.006 ^e^	83.6 ^b^
13	4.0	4.0	0.161 ± 0.012 ^e^	86.9 ^ab^

^1^ *L. rhamnosus* (1.0 × 10^10^ cells/g) and *S. cerevisiae* (1.0 × 10^10^ cells/g) biomasses inactivated by autoclaving at 121 °C for 10 min. ^2^ Values are expressed as mean ± standard deviation of samples analyzed in triplicate. ^3^ Percentage reductions relative to aflatoxin B_1_ mean concentration in controls. ^a–f^ In the same column, mean values followed by different superscript letters differ significantly (*p* < 0.05).

**Table 2 foods-13-03299-t002:** Physicochemical characteristics of yogurts incorporated with different percentages of inactivated biomasses ^1^ of *Lacticaseibacillus rhamnosus* and *Saccharomyces cerevisiae*, on day 30 of storage at 5 °C.

Treatment (Vat)	*L. rhamnosus* (%)	*S. cerevisiae* (%)	Fat (%) ^2^	Protein (%) ^2^	pH ^2^
1 (Control)	0	0	1.05 ± 0.02	4.11 ± 0.03	4.47 ± 0.01
2	0.5	0	1.03 ± 0.05	4.13 ± 0.02	4.53 ± 0.03
3	0	0.5	1.06 ± 0.03	4.20 ± 0.05	4.52 ± 0.01
4	0.5	0.5	1.04 ± 0.04	4.08 ± 0.04	4.49 ± 0.02
5	1.0	0	1.07 ± 0.04	4.09 ± 0.08	4.58 ± 0.03
6	0	1.0	1.09 ± 0.03	4.26 ± 0.07	4.52 ± 0.03
7	1.0	1.0	1.03 ± 0.01	4.18 ± 0.03	4.49 ± 0.02
8	2.0	0	1.05 ± 0.04	4.17 ± 0.01	4.46 ± 0.01
9	0	2.0	1.08 ± 0.03	4.23 ± 0.08	4.54 ± 0.02
10	2.0	2.0	1.07 ± 0.02	4.14 ± 0.03	4.47 ± 0.05
11	4.0	0	1.04 ± 0.02	4.12 ± 0.07	4.50 ± 0.03
12	0	4.0	1.01 ± 0.04	4.20 ± 0.01	4.52 ± 0.03
13	4.0	4.0	1.05 ± 0.03	4.10 ± 0.01	4.48 ± 0.02

^1^ *L. rhamnosus* (1.0 × 10^10^ cells/g) and *S. cerevisiae* (1.0 × 10^10^ cells/g) biomasses inactivated by autoclaving at 121 °C for 10 min. ^2^ Values are expressed as mean ± standard deviation of samples analyzed in triplicate. No significant differences (*p* < 0.05) were found between mean values in the same column.

**Table 3 foods-13-03299-t003:** Sensory characteristics ^1^ of yogurts incorporated with different percentages of inactivated biomasses ^2^ of *Lacticaseibacillus rhamnosus* and *Saccharomyces cerevisiae*, on day 30 of storage at 5 °C.

Treatment (Vat)	*L. rhamnosus* (%)	*S. cerevisiae* (%)	Appearance	Consistency	Aroma	Taste
1 (Control)	0	0	3.6 ± 0.8	3.8 ± 0.8	2.9 ± 0.7	3.0 ± 1.0
2	0.5	0	3.8 ± 0.4	3.8 ± 0.4	2.9 ± 0.7	2.6 ± 0.4
3	0	0.5	3.5 ± 0.5	4.0 ± 0.7	3.3 ± 0.4	3.0 ± 0.7
4	0.5	0.5	3.3 ± 0.8	3.5 ± 0.9	2.6 ± 0.4	3.2 ± 0.9
5	1.0	0	3.8 ± 0.4	3.8 ± 0.4	3.2 ± 0.5	2.8 ± 0.3
6	0	1.0	3.5 ± 0.5	3.8 ± 0.8	3.4 ± 0.6	3.3 ± 0.8
7	1.0	1.0	3.5 ± 0.5	3.6 ± 1.0	2.9 ± 0.2	3.1 ± 0.6
8	2.0	0	3.8 ± 0.4	3.8 ± 0.4	3.0 ± 0.1	2.8 ± 0.4
9	0	2.0	3.8 ± 0.4	3.9 ± 0.9	3.5 ± 0.6	3.6 ± 0.5
10	2.0	2.0	3.3 ± 0.4	3.3 ± 0.8	2.6 ± 0.4	2.8 ± 0.8
11	4.0	0	3.5 ± 0.5	3.8 ± 0.4	2.8 ± 0.5	3.2 ± 0.3
12	0	4.0	3.5 ± 0.9	4.0 ± 0.7	3.1 ± 0.5	3.0 ± 0.1
13	4.0	4.0	3.3 ± 0.4	3.3 ± 0.8	3.1 ± 0.7	3.3 ± 0.4

^1^ Values are expressed as mean ± standard deviation of individual scores obtained from 15 trained panelists. ^2^ *L. rhamnosus* (1.0 × 10^10^ cells/g) and *S. cerevisiae* (1.0 × 10^10^ cells/g) biomasses inactivated by autoclaving at 121 °C for 10 min. No significant differences (*p* < 0.05) were found between mean values displayed in the same column.

## Data Availability

The original contributions presented in the study are included in the article, further inquiries can be directed to the corresponding author.

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
