# Peer review of "Evaluation of Ability of Inactivated Biomasses of Lacticaseibacillus rhamnosus and Saccharomyces cerevisiae to Adsorb Aflatoxin B1 In Vitro"

_foods, 2024, doi:10.3390/foods13203299_

Round 1
Reviewer 1 Report
Comments and Suggestions for Authors
The manuscript entitled “Evaluation of Biomasses of Lacticaseibacillus rhamnosus and Saccharomyces cerevisiae to Adsorb Aflatoxin B1 In Vitro” aims evaluate the efficacy of inactivated biomasses of Lacticaseibacillus rhamnosus and Saccharomyces cerevisiae, alone or in combination, as components in functional yogurts to adsorb aflatoxin B1 in vitro. The study further investigates the impact of these biomasses on aflatoxin adsorption, cell survival, and the physicochemical and sensory properties of the yogurts, exploring the potential of such yogurts as a strategy for reducing dietary exposure to aflatoxin B1.
The manuscript is well prepared in general. It is in the scope of the journal. My minor comments are given below.
The abstract ends abruptly without discussing the broader implications of the findings, such as the practical application or scalability of using these biomasses in functional foods to mitigate aflatoxin exposure.
Line 133: The procedure should be described in detail for the sake of completeness.
Sensory Evaluation of Functional Yogurts: the authors should explain why they chose a male group of 15 individuals. Does that allow for the proper statistical analysis?
The conclusion should clearly restate the specific adsorption percentages achieved in the study, as this was a significant result. Also, it should address the limitations of the study and some future directions.
Comments on the Quality of English Language
Minor changes are required.
Author Response
Reviewer #1:
The manuscript entitled “Evaluation of Biomasses of Lacticaseibacillus rhamnosus and Saccharomyces cerevisiae to Adsorb Aflatoxin B1 In Vitro” aims evaluate the efficacy of inactivated biomasses of Lacticaseibacillus rhamnosus and Saccharomyces cerevisiae, alone or in combination, as components in functional yogurts to adsorb aflatoxin B1 in vitro. The study further investigates the impact of these biomasses on aflatoxin adsorption, cell survival, and the physicochemical and sensory properties of the yogurts, exploring the potential of such yogurts as a strategy for reducing dietary exposure to aflatoxin B1.
The manuscript is well prepared in general. It is in the scope of the journal. My minor comments are given below.
Answer: Thanks for the positive comments.
The abstract ends abruptly without discussing the broader implications of the findings, such as the practical application or scalability of using these biomasses in functional foods to mitigate aflatoxin exposure.
Answer: Thanks for the suggestion. A sentence was included at the end of the Abstract, to provide the required information (please see L.35-37).
Line 133: The procedure should be described in detail for the sake of completeness.
Answer: The description of procedures was improved, as requested (kindly see L.111-121).
Sensory Evaluation of Functional Yogurts: the authors should explain why they chose a male group of 15 individuals. Does that allow for the proper statistical analysis?
Answer: This was wrongly mentioned in the original manuscript. In fact, the trained panel was formed by 8 men and 7 women, totaling 15 individuals as amended in the revised manuscript (please see L.234-235).
The conclusion should clearly restate the specific adsorption percentages achieved in the study, as this was a significant result. Also, it should address the limitations of the study and some future directions.
Answer: The adsorption percentages, the limitations of the study and prospects were included in the Conclusion section, as requested (please see L.398-400, 405-414 and L.416-422, respectively).
Reviewer 2 Report
Comments and Suggestions for Authors
1. The topic of the work is interesting. Although there are already many publications proving the effectiveness of microorganisms in eliminating mycotoxins
2. In my opinion, not all aspects of the work are obvious so they need explanation
The aim of the work is clearly defined. However, it does not explain why the authors studied yogurt. Please indicate in the introduction of the work what this results from and what the scale of the problem is (contamination of yogurts with mycotoxins). Reading the publication, it was not clear whether the goal was to eliminate the toxin from the product as a secondary contaminated matrix (in this case, yogurt) or to eliminate the toxin introduced from the raw material (milk). Moreover, please refer to the impact of yogurt production technology on the possible elimination/transformation of the toxins contained.
3. Introduction
Ver. 49: the context suggests that probiotic bacteria are LAB. This is not true. Probioticity is a strain characteristic and most probiotic bacteria belong to the metabolic group of lactic fermentation bacteria. The statement should be clarified. I have a similar comment about yeasts – please clarify the statement. In this form, it may be misleading to the reader.
Definition of postbiotics. The indicated definition may mislead the reader. The reviewer should not suggest literature that should be used in the work. There are, however, publications that describe the basic differences between “postbiotics” and “parabiotics”. Please adapt the content to these definitions
Please enter in the table description that the results refer to products stored for 30 days
The methodology does not provide information on how the presence of L.r and S.c. in the yogurt samples shown in the tables was determined
In my opinion, the discussion of the results is not exhaustive. For example, there is no explanation of the mechanisms of detoxification by the microorganisms used.
4. An interesting aspect of the work is the suggestion of the possibility of using postbiotics in yogurt (ex. as nutridrink) to mitigate the effects of exposure to AF - ver. 252 and 253.
Author Response
Reviewer #2:
- The topic of the work is interesting. Although there are already many publications proving the effectiveness of microorganisms in eliminating mycotoxins
Answer: Thanks for the comments. Yes, there are many publications proving the effectiveness of microorganisms in eliminating mycotoxins. However, our work is the first study describing the protective effects of functional yogurts containing inactivated biomasses of microorganisms against the cytotoxic effects of AFB1, and the potential influence of the incorporation of these products on the quality parameters of yogurts, as amended in L.379-388 of the revised manuscript.
- In my opinion, not all aspects of the work are obvious so they need explanation
The aim of the work is clearly defined. However, it does not explain why the authors studied yogurt. Please indicate in the introduction of the work what this results from and what the scale of the problem is (contamination of yogurts with mycotoxins). Reading the publication, it was not clear whether the goal was to eliminate the toxin from the product as a secondary contaminated matrix (in this case, yogurt) or to eliminate the toxin introduced from the raw material (milk). Moreover, please refer to the impact of yogurt production technology on the possible elimination/transformation of the toxins contained.
Answer: Yogurt is one of the most common probiotic-enriched foods, which explains why the authors studied this product in the experiment. This was amended in the revised manuscript, please see L.75-79. Regarding the other comments in this item, it seems there has been some misunderstanding: The results presented are not related to the contamination of yogurts with mycotoxins, and the goal was not to eliminate the toxin from the product as a secondary contaminated matrix (in this case, yogurt), neither to eliminate the toxin introduced from the raw material (milk). As you kindly mentioned in the beginning of this item (“The aim of the work is clearly defined”), the study aimed to evaluate the efficacy of inactivated biomasses of L. rhamnosus and S. cerevisiae, incorporated alone or in combination into plain, non-flavored yogurts as vehicles for these inactivated microorganisms, to adsorb AFB1 in vitro and reduce its toxicity on MEF-1 cells. Therefore, we did not assess the possible impact of yogurt production technology on the elimination/transformation of the toxins contained, because the aflatoxin (AFB1) was not incorporated into milk or yogurt during its manufacture but added to the yogurt after its manufacture and incorporation of the inactivated biomasses in test tubes (for the adsorption assay) and on MEF-1 cell in culture plates (for the cytotoxicity assay).
- Introduction
Ver. 49: the context suggests that probiotic bacteria are LAB. This is not true. Probioticity is a strain characteristic and most probiotic bacteria belong to the metabolic group of lactic fermentation bacteria. The statement should be clarified. I have a similar comment about yeasts – please clarify the statement. In this form, it may be misleading to the reader.
Answer: The information regarding the probiotic microorganisms was corrected for clarity, please see L.69-73.
Definition of postbiotics. The indicated definition may mislead the reader. The reviewer should not suggest literature that should be used in the work. There are, however, publications that describe the basic differences between “postbiotics” and “parabiotics”. Please adapt the content to these definitions
Answer: Thanks for highlighting this issue. The term “postbiotics” was replaced with the correct term, “paraprobiotics’, throughout the manuscript, with a proper reference to support their definition and properties (Lee et al. 2023 [14]).
Please enter in the table description that the results refer to products stored for 30 days
Answer: The information requested was included in Table 1, as requested (please see L.272).
The methodology does not provide information on how the presence of L.r and S.c. in the yogurt samples shown in the tables was determined
Answer: There has been some misunderstanding here too: We did not use viable cells of L.r or S.c., only their inactivated biomasses. They were incorporated into yogurt with increasing percentages (w/w) in the yogurt-producing vats (kindly see L.136-141 of the revised manuscript). So, there is no point in determining the presence of L.r or S.c. in the yogurt samples.
In my opinion, the discussion of the results is not exhaustive. For example, there is no explanation of the mechanisms of detoxification by the microorganisms used.
Answer: The discussion was improved by adding the known mechanisms behind the adsorption process of AFB1 by LAB and yeast cells (please see L.261-270).
- An interesting aspect of the work is the suggestion of the possibility of using postbiotics in yogurt (ex. as nutridrink) to mitigate the effects of exposure to AF - ver. 252 and 253.
Answer: Thanks for the comment.
Reviewer 3 Report
Comments and Suggestions for Authors
The manuscript entitled: “Evaluation of Biomasses of Lacticaseibacillus rhamnosus and Saccharomyces cerevisiae to Adsorb Aflatoxin B1 In Vitro” reports a in vitro study focused on the use of inactivated biomasses of Lacticaseibacillus rhamnosus and Saccharomyces cerevisiae used by alone or in combination added to functional yogurts to adsorb aflatoxin B1. The decontamination of secondary metabolites from Aspergillus species triggering interest in the research within this area of interest and it adds information to the area of interest.
Nonetheless, the scope of the proposed submitted manuscript and the overall novelty of the approach described should be better defined and detailed compared to studies in the area of interest. In particular, it is mentioned “adsorb Aflatoxin B1 In Vitro”: please specify better which section of the manuscript details this point.
Some other comments are reported in the following.
The title of the manuscript should also include “inactivated” for clarity of the context. The section 2.1. Preparation of Lactic Acid Bacteria and Yeast Biomasses” should contan more dtails experimental about biomassess mentioned. Comparison with alternative detoxification/decontaminatio options would be useful to assess the context.
Please specify better the meaning of “plain yogurts” at line 99 indicating the origin and how it was obtained linking better this part to the following section of the manuscript.
At line 104 please specify the amounts avoiding generic statements (e.g. :“convenient amounts”).
To mention “non-detectable levels” should be avoided: please refer to the LOD and LOQ. Why samples have been evaluated in duplicate? Please comment this point.
Section 2.3 experimental procedures should be better described and in a clearer way. The results and conclusion seems to need a revision for easier readability and assessing of the reported results as in the Tables.
The criterium for the selection of the panelists for the sensory analysis should be better justified (only men have been involved?): please explain better and motivate. The etical committee approval should be mentioned.
Please explain better the correlation between the sensory results and the effect on the studied cell lines. The possible impact mentioned in the Conclusion section of the manuscript on the “preventive strategy for dietary exposure to aflatoxins” should be better substantiated. The possible impact of the proposed approach and methodology should be better overall substantiated.
Comments on the Quality of English LanguageModerate English language check could be needed for better clarity and readability.
Author Response
Reviewer #3:
The manuscript entitled: “Evaluation of Biomasses of Lacticaseibacillus rhamnosus and Saccharomyces cerevisiae to Adsorb Aflatoxin B1 In Vitro” reports a in vitro study focused on the use of inactivated biomasses of Lacticaseibacillus rhamnosus and Saccharomyces cerevisiae used by alone or in combination added to functional yogurts to adsorb aflatoxin B1. The decontamination of secondary metabolites from Aspergillus species triggering interest in the research within this area of interest and it adds information to the area of interest.
Answer: Thanks for the positive comments.
Nonetheless, the scope of the proposed submitted manuscript and the overall novelty of the approach described should be better defined and detailed compared to studies in the area of interest. In particular, it is mentioned “adsorb Aflatoxin B1 In Vitro”: please specify better which section of the manuscript details this point.
Answer: Our manuscript evaluates a combination of inactivated microorganisms in yogurt (used as a vehicle) to adsorb AFB1 in vitro. Most of the studies using yogurt as a vehicle for beneficial microorganisms are performed with live isolated cells and, although they possess adequate properties, they may change the physicochemical and sensory characteristics of the products. The utilization of inactivated microbial cells has been studied mainly using single or combined microorganisms in aqueous solutions, not incorporated into food matrices. In this trial, the results indicated for the first time that the addition of L. rhamnosus and S. cerevisiae biomasses to yogurts was effective to adsorb AFB1 and reduce its toxicity on MEF-1 cells, when simultaneously exposed to the functional yogurts containing ³ 1.0% (w/w) of biomasses. In addition, no significant difference was noted in the physicochemical characteristics and sensory attributes between the aflatoxin-free, control and functional yogurts with biomasses at inclusion percentages of up to 4.0% (w/w) during 30 days of storage, which also highlights the novelty of the study. This information was amended in the revised manuscript, please see L.379-388.
Section 2.3 was amended with the term “in vitro”, to specify better which section of the manuscript details this point, as suggested (kindly see L.151).
Some other comments are reported in the following.
The title of the manuscript should also include “inactivated” for clarity of the context. The section 2.1. Preparation of Lactic Acid Bacteria and Yeast Biomasses” should contan more dtails experimental about biomassess mentioned. Comparison with alternative detoxification/decontaminatio options would be useful to assess the context.
Answer: The title was modified, and section 2.1 was improved, as requested (please see L.2 and L.109-121, respectively).
Please specify better the meaning of “plain yogurts” at line 99 indicating the origin and how it was obtained linking better this part to the following section of the manuscript.
Answer: Plain yogurt refers to the non-flavored product (without any addition of flavor components, including fruits, jams, or sugar), according to Soukoulis et al. 2007, as amended in the revised manuscript (please see L.103).
At line 104 please specify the amounts avoiding generic statements (e.g. :“convenient amounts”).
Answer: Done (please see L.129).
To mention “non-detectable levels” should be avoided: please refer to the LOD and LOQ. Why samples have been evaluated in duplicate? Please comment this point.
Answer: The information regarding the detection limits for AFB1 and AFM1 was included in the new, revised version of the manuscript (please see L.135-136). Samples of the standardized milk were analyzed in triplicate (not duplicate), to reduce the experimental error and increase the accuracy of the analytical results.
Section 2.3 experimental procedures should be better described and in a clearer way. The results and conclusion seems to need a revision for easier readability and assessing of the reported results as in the Tables.
Answer: Section 2.3 was revised to improve the description of the experimental procedures (please see L.152-158 and L.175-176). The Results & discussion and Conclusion sections were improved and revised for clarity, as suggested.
The criterium for the selection of the panelists for the sensory analysis should be better justified (only men have been involved?): please explain better and motivate. The etical committee approval should be mentioned.
Answer: This was wrongly mentioned in the original manuscript. In fact, the trained panel was formed by 8 men and 7 women, totaling 15 individuals, as amended in the revised manuscript (please see L.234-235). The ethical committee approval was included, as requested (please see L.235-236).
Please explain better the correlation between the sensory results and the effect on the studied cell lines. The possible impact mentioned in the Conclusion section of the manuscript on the “preventive strategy for dietary exposure to aflatoxins” should be better substantiated. The possible impact of the proposed approach and methodology should be better overall substantiated.
Answer: The cell lines (MEF-1 cells) were used in the experiment to evaluate the ability of yogurts containing biomasses to reduce the cytotoxicity of aflatoxin B1. So, no correlation is expected between the sensory results and the effect on the studied cell lines. The possible impact mentioned in the Conclusion section and the proposed approach and methodology was better substantiated, as suggested (please see L.415-422).